# Refractometric Sensitivity Enhancement of Weakly Tilted Fiber Bragg Grating Integrated with Black Phosphorus

**DOI:** 10.3390/nano10071423

**Published:** 2020-07-21

**Authors:** Zhao Zhang, Kun Liu, Junfeng Jiang, Tianhua Xu, Shuang Wang, Jinying Ma, Pengxiang Chang, Jiahang Zhang, Tiegen Liu

**Affiliations:** 1School of Precision Instruments and Opto-electronics Engineering, Tianjin University, Tianjin 300072, China; zhangzhao0905@tju.edu.cn (Z.Z.); xutianhua@tju.edu.cn (T.X.); shuangwang@tju.edu.cn (S.W.); majinying@tju.edu.cn (J.M.); pxchang@tju.edu.cn (P.C.); wuwen@tju.edu.cn (J.Z.); tgliu@tju.edu.cn (T.L.); 2Key Laboratory of Opto-electronics Information Technology, Ministry of Education, Tianjin 300072, China; 3Tianjin Optical Fiber Sensing Engineering Center, Institute of Optical Fiber Sensing of Tianjin University, Tianjin 300072, China

**Keywords:** weakly tilted fiber Bragg, black phosphorus, refractometric sensing

## Abstract

The sensitivity enhancement of the weakly tilted fiber Bragg grating (WTFBG) integrated with black phosphorus (BP) was investigated via numerical simulations and experimental demonstrations. BP nanosheets were deposited twice on the cylindrical WTFBG surface using the in situ layer-by-layer (i-LbL) deposition technique. The resonance intensity of the deepest cladding mode located around 1552 nm of WTFBG had a 9.2 dB decrease after the BP deposition process. This allows for the application of the intensity-modulated refractive index (RI) sensor. The sensing platform was implemented on the use of the BP integrated with WTFBG (BP-WTFBG). The refractometric sensing was achieved with the sensitivity enhancement of the resonance intensity modulation of the deepest cladding mode for the BP-WTFBG. The sensitivities were 137.6 dB/RIU and 75.6 dB/RIU in the RI region of 1.33–1.35 and 1.35–1.38, respectively. This platform shows great potential applications for biochemical sensing because of its highly sensitive RI sensing ability around the biochemical sensing window.

## 1. Introduction

At present, research into optical fiber based refractive index (RI) sensing has rapidly developed the areas of biochemical sensing, environmental monitoring, and life science due to well-known advantages such as compact size, immunity to electromagnetic interference, real-time sensing, and so on [1,2,3]. The fiber Bragg grating (FBG), consisted of short- or long-period fiber gratings, can achieve multi-parameter and in-line measurements [4,5,6,7,8,9]. A special type of short-period FBG, called the weakly tilted fiber Bragg grating (WTFBG), can be inscribed with a small tilt angle in the single mode fiber core, where no secondary modification such as etched, tapered, or side-polished operation is required. A series of cladding modes in the WTFBG are excited, which are sensitive to the change of surrounding environments. The WTFBG is a feasible choice for the detection of physical and biochemical quantities (e.g., the surrounding refractive index (SRI)) [10,11,12]. However, the wavelength demodulation based on the bare WTFBG has a very low sensitivity in the region of low refractive index, and this constrains its application in the areas of biochemical sensing [13]. To improve the sensitivity in the low SRI region (such as water solution), some trials that have increased the tilted angle [4] and coated the surface with the metal film to excite the surface plasmon resonance (SPR) to enhance the sensing ability using P-polarized light [3,14] have been carried out to improve the sensitivity of WTFBG in the low SRI region (e.g., in the water solution). However, it is difficult to demodulate the wavelength envelope of SPR in the metal coated WTFBG. On the other hand, the WTFBG integrated with black phosphorus (BP) offers a solution to improve the sensitivity and simplify the demodulation of the WTFBG sensing system, where an intensity modulation is applied.

Black phosphorus, as a branch of famous two-dimensional (2D) materials, has shown characteristics of higher surface-to-volume ratio and carrier mobility compared to other 2D materials. Due to the aforementioned superiorities, BP has been applied in BP field transistors (FETs), photodetectors, and chemical sensors [15,16,17]. Recently, a branch of long-period gratings was integrated with BP where the product was applied to detect heavy metal ions [18]. However, this FBG requires high laser power and stricter optical alignment conditions in the inscription process, which leads to obstacles in the fabrication, leading to many difficulties in the fabrication process. In addition, the cladding modes of the ETFBG has a larger full width at half maximum (FWHM, ~4–5 nm) [19,20] compared to that of the WTFBG (~0.2 nm) due to different mechanisms of cladding coupling. The WTFBG employs backward coupling while the ETFBG uses forward coupling [4]. In principle, the WTFBG will provide higher flexibility and sensitivity in both the implementation and application of the biochemical sensing platform.

In this paper, we placed BP layers onto a WTFBG using the in situ layer-by-layer (i-LbL) deposition approach to produce a novel sensing platform, where sensitivity enhancement of the light–matter interaction and refractive index detection were achieved. Due to the interaction between the BP coating and the cladding modes of the WTFBG, a significant change in the intensity of the cladding mode resonance was observed compared to that in the bare WTFBG. The BP coated WTFBG sensor showed significantly higher sensitivity and dynamic range for detecting the refractive indices of different solutions when the intensity-demodulated cladding mode approach was applied.

## 2. Theory, Simulations, Materials, and Methods

### 2.1. Theory

The short period WTFBG with a tilted angle of 8° ± 1° was inscribed by using an ultraviolet (UV) laser in hydrogen loaded standard SM-28 fiber with a mask, the tilted angle of which is similar to the WTFBG. The light propagating in the WTFBG follows the phase-matching theory, which provides the wavelength position of the resonance band according to the coupling between two modes. The resonance wavelength of the core mode $\lambda_{Bragg}$ and the wavelength at the ith cladding mode $\lambda_{cl,i}$ are given by [21]
(1)$$\lambda_{\mathrm{Bragg}}=2n_{eff,core}\frac{\Lambda }{\cos\theta }$$
(2)$$\lambda_{cl,i}=(n_{eff,core}+2n_{eff,cl,i})\frac{\Lambda }{\cos\theta }$$
where $n_{eff,core}$ and $n_{eff,cl,i}$ are the effective index of the fiber core and ith cladding mode, respectively. Λ is the grating period and θ is the tilted angle between the grating plane and the vertical line of the fiber.
(3)$$R=\tanh^{2}(\kappa L)$$

The resonance strength (also called depth) of the cladding modes (their power reflectivity *R*) strongly depends on the coupling coefficient κ between the core mode and the cladding modes, and also significantly relies on the length *L* of the grating region according to Equation (3) [22]. The coupling coefficient κ is expressed as
(4)$$\kappa =C\int \int_{-\infty }^{\infty }\vec{E_{core}}\cdot \vec{E_{r}}\Delta n(x,y)dxdy$$

It can be seen that the coupling efficient κ is determined by transverse components in the electric fields of the considered modes [22]. C is a proportionality constant related to the transverse mode fields ($E_{core}$ and $E_{r}$). $E_{core}$ and $E_{r}$ are intensities of the electric fields in the core and the high-order cladding modes, respectively. Δn(x,y) describes the perturbation of the refractive index due to the grating structure in the fiber cross-section. In our experiment, Δn(x,y) mainly depends on the change in the effective index of the BP film.

### 2.2. Simulations

The working principle of RI sensing of bare WTFBG is governed by the phase-matching condition as shown in Equation (2). However, when the RI of the external medium is higher than the fiber cladding, the phase matching cannot be satisfied. The external medium of higher RI than the fiber cladding will result in a loss of the total internal reflection condition of the light guided by the cladding for the WTFBG. These cladding modes will then become radiation modes (leaky cladding modes). A portion of the cladding modes will be reflected at the cladding/external medium interface while the rest will be transmitted and lost to the external medium. The amount of reflectance affects the resonance intensity of the cladding modes based on the Fresnel reflection coefficients.

In our scheme, the BP film, which has a higher RI than the fiber cladding, is deposited on the surrounding of the grating region of the WTFBG. The system can be modified to incorporate the BP film as the additional dielectric layer. To verify the effectiveness of the BP, a four layer fiber cylindrical waveguide structure (fiber core-fiber cladding-BP film-external medium) model was developed, as shown in Figure 1. The amount of reflectance of the cladding mode at the fiber cladding/BP film interface can be determined by [23].
(5)$$R={\left|\frac{r_{2,3}+r_{3,4}\exp(-i{\tilde{k}}_{film})}{1+r_{2,3}r_{3,4}\exp(-i{\tilde{k}}_{film})}\right|}^{2}$$
(6)$$r_{2,3}=\frac{n_{clad}-{\tilde{n}}_{film}}{n_{clad}+{\tilde{n}}_{film}}$$
(7)$$r_{3,4}=\frac{{\tilde{n}}_{film}-n_{ext}}{{\tilde{n}}_{film}+n_{ext}}$$
(8)$${\tilde{k}}_{film}=\frac{4\pi {\tilde{n}}_{film}d_{film}}{\lambda }=\frac{4\pi nd_{film}}{\lambda }-i\frac{4\pi kd_{film}}{\lambda }=\beta_{film}-iad_{film}$$
$$a=\frac{4\pi }{\lambda }\;\;\;{\tilde{n}}_{film}=n-ik$$
where a is the absorption coefficient of the BP film; $n_{clad}$ and $n_{ext}$ are the RIs of the fiber cladding and the external medium, respectively. d is the thickness of the BP film. $r_{x,y}$ is the amplitude of the reflection coefficient of the interface between layer x and layer y, as shown in Figure 1. From Equations (5)–(8), we can see that R would vary with $n_{ext}$ and will affect the depth of attenuation of WTFBG. Therefore, the reflectance R is related to the depth of resonance intensity of the cladding modes of the WTFBG.

According to Figure 2, it is seen that the BP film has obvious optical absorption phenomenon when compared with the uncoated structure. With the RI increased, the reflectance intensity of the fiber coated with BP decreased, correspondingly. However, there was no optical absorption effect in the bare fiber. The phenomenon caused originates from the complex refractive index and the strong optical absorption property of the BP. Both characteristics can change the boundary condition of the fiber, leading to the leakage of light from the TFBG cladding to the BP coating. This phenomenon also extends the interaction space of cladding modes.

### 2.3. Materials and Methods

BP nanosheets were produced by the liquid-phase exfoliation method and were then immersed into pure isopropyl alcohol (IPA) with a concentration of 0.5 mg/mL. The number of layers used in our experiments were 1–10 layers and the diameter of the nanosheets was about 100 nm–5 μm.

The BP deposition was implemented based on the modification of the chemical surface using cascaded in situ layers, according to the i-LbL self-assembly technique. The main idea of the layer-by-layer (LbL) method consists of the assembly of oppositely electrically charged particles that form a bilayer; the process can be repeated as many times as the thickness of the film requires. However, the surface of bare fiber is uncharged, and some functionalization for the fiber surface needs to be conducted to make the fiber charged. Flow diagrams are illustrated in Figure 3. First, the WTFBG surface was cleaned with pure acetone to remove contaminants. Second, the fiber hydroxylation was carried out by immersing the fiber with 1.0 M NaOH aqueous solution for 1 h at room temperature to enrich the number of –OH group on the fiber surface. After that, the fiber was washed with deionized water and ethanol three times consecutively to remove the redundant hydroxyl. Third, the fiber silanization was performed using a fresh 5% 3-aminopropyltriethoxysilane (APTES) ethanol solution (with a reaction for 3 h) to form the Si–O–Si–NH_2_ chemical bond on the fiber surface at room temperature. Then, the fiber was cleaned with ethanol to remove the unbonded monomers and then placed in a drying oven at 70 °C for 30 min to stabilize the APTES adhesion. Finally, the fiber was fixed into a homemade V-Type microchannel container with the 60 μL BP solution carefully added and then the fiber was heated at 50 °C, leading to a better combination with the BP nanosheets. Negatively-charged BP nanosheets can incorporate closely with positively-charged amino groups on the APTES-silanized fiber surface via electrostatic force. The second-round deposition started when the solvent of the BP solution had fully evaporated. After this, the BP-coated WTFBG was cleaned again using the pure ethanol to remove unbonded BP nanosheets, and was further dried and stored in a vacuum environment to enhance the adhesion of BP on the fiber surface.

The surface morphology of BP-WTFBG was characterized using an optical microscope, Raman spectroscopy (InVia Reflex, RENISHAW.Inc, Gloucestershire, England), and scanning electron microscopy (SEM) using a JSM7610F microscope (HITACHI, Tokyo, Japan), respectively. The optical microscope image in Figure 4a shows that the BP nanosheets were well coated on the fiber surface. Meanwhile, three Raman spectra were applied to examine the coating material including $A_{g}^{1}$(362 cm^−1^), $B_{2g}$(438 cm^−1^), and $A_{g}^{2}$(466 cm^−1^), which were clearly observed in Figure 3b, indicating the presence of black phosphorus [24]. The surface morphology of the BP coated WTFBG was also detected by SEM (Figure 4c,d) with magnifications of 800× and 5000×, respectively. It was found that the BP nanosheets were closely and densely attached to the cylindrical surface of the fiber, proving the high quality of the BP deposition.

## 3. Experiments and Results

### 3.1. Experimental Setup

The refractive index solutions, which consisted of different proportions of deionized water and glycerol with a RI range from 1.33 to 1.38, were detected using the BP-WTFBG sensor to evaluate the sensing ability. The schematic structure of the optical detection platform is shown in Figure 5. The line-polarized light, which came from the tunable laser scanning system (TLSS, 8164B, Agilent Technologies Inc, San Diego, CA, USA), was adjusted by a three-ring polarization controller (PC) to change the polarization state and then fed into the BP-WTFBG or the bare WTFBG sensor. Both sensors were immersed in different RI solutions for SRI detection. The transmission spectra of the BP-WTFBG or bare WTFBG sensor were collected by the power input port of the TLSS and then the spectrum data were analyzed using a laptop. The cleaning process needs to be carried out by using deionized water three times before next detection to ensure the accuracy of the detection results. During the experiments, the temperature was kept at 25 ± 0.5 °C.

### 3.2. The Variation of Optical Transmission Spectra for Weakly Tilted Fiber Bragg Grating (WTFBG) with Black Phosphorus (BP) Deposition

The transmission spectrum of the WTFBG changed gradually in accordance with the BP deposition process. For the 8° tilted WTFBG, the tilted plane grating breaks the symmetry of mode coupling. In Figure 6, the variation of the transmission spectrum of the WTFBG with double BP depositions can be observed. From Figure 6b, the resonance strength of the cladding mode around 1552 nm presented a 9.2 dB decrease. This phenomenon arises from the strong optical absorption of the BP, which changes the electric intensity of the cladding mode and further affects the coupling efficiency between the core mode and the cladding mode, according to Equation (4). The wavelength red-shift for the cladding mode was attributed to the increase in the cladding effective index, which comes from the high RI of the BP coating layer, according to phase-matching condition shown in Equation (2).

### 3.3. Refractometric Response Measurement for Bare and BP Coated with WTFBG

The refractometric response of the two types of WTFBGs (with and without BP film) were compared from the transmission spectra in the detection of solutions with different refractive indices.

Figure 7a,c shows the results of RI measurements using the BP-WTFBG sensor. The resonance intensity of the cladding modes, especially the deepest one (Figure 7c), showed an obvious decrease with the increase of RI. These can prove that the reflectance of the cladding modes at the cladding/BP film interface is decreased due to the presence of BP film. For the comparative group, the bare WTFBG’s transmission spectrum is shown in Figure 7b,d, and the intensity tendency of the deepest cladding mode illustrated in Figure 7a shows that it was almost stable in the same RI region. This can also prove that the BP has the ability of high optical absorption and optical modulation.

Figure 8a shows the variation of resonance strength of the deepest cladding for BP-coated/bare WTFBG, where the BP-WTFBG had an intensity modulation but the bare WTFBG did not appear. The results are consistent with the simulation in Section 2.1, as shown in Figure 2. The sensitivity of BP-WTFBG reached 137.6 dB/RIU and 75.6 dB/RIU for the RI range from 1.33 to 1.35 (R^2^ = 9995) and 1.35 to 1.38 (R^2^ = 9972), as shown in Figure 8b, respectively, with excellent linear tendency. Compared to the simulation results, different sensitivities in the two RI regions may arise from the lack of the compactness in the BP deposition or the nonlinearity of BP’s complex refractive index under different wavelengths. In particular, the sensitivity of BP-WTFBG in the 1.33–1.35 RI region showed that it was 4.4 times, 2.5 times, and 1.8 times higher than those in the long period fiber grating coated carbon-nanotube [25], microfiber Bragg grating coated with carbon nanotube [26], and graphene oxide functionalized long period fiber grating [27], respectively. Moreover, due to the significantly smaller FWHM of the cladding mode (0.1–0.2 nm), the BP-WTFBG sensor had a higher figure of merit (FOM) when compared with ETFBG (25 times higher) and LPG (250 times higher) at a similar sensitivity level.

### 3.4. Discussion

From the aforementioned results, the BP-WTFBG had an obvious sensitivity enhancement for the solution with low RIs when intensity-demodulated of cladding mode was applied. Two factors played a dominant role. The first is the characteristics of BP such as excellent large surface-to-volume ratio, high molecule adsorption ability, and outstanding optical absorption property at the near infrared wavelength. Another is the modulated distribution of the evanescent field intensity because of the complex RI of the coated BP. Therefore, the BP film lead the attenuation strength of the WTFBG’s cladding modes to be more sensitive to the external environments.

From the above analysis, the proposed BP-WTFBG sensor shows a significant sensitivity enhancement of the low-RI region, which involves a biochemical sensing window of biochemical sensing. In addition, the BP-WTFBG provides a low-cost, harmless (to human body) and environmentally-friendly solution for biochemical sensing.

## 4. Conclusions

In this paper, the application of a 2D material, BP, in the WTFBG sensing system, was investigated considering its optical absorption properties at the near-infrared region. Numerical simulations were performed to prove the high optical absorption of the BP coating on the fiber, BP nanosheets were deposited onto the cylindrical fiber based on the i-LbL deposition technique, and a new sensing system was developed based on the use of BP coated WTFBG. The surface morphology and characteristics of BP film were examined by SEM and Raman spectroscopy, which showed that the BP nanosheets had good attachment on the fiber surface. High-sensitivity measurements of aqueous solution with low RIs were experimentally carried out using the BP-WTFBG sensor. Compared to the bare WTFBG, which had no intensity response of cladding modes against the change in the RI, the sensitivity of BP-WTFBG platform was 137.6 dB/RIU and 75.6 dB/RIU in the RI region of 1.33–1.35 and 1.35–1.38, respectively. The experimental results showed a good agreement with the numerical simulations. The RI detection using the proposed BP-WTFBG sensor showed a significantly higher sensitivity than the use of the LPG coated with the carbon-nanotube, the microfiber Bragg grating coated with the carbon-nanotube, and the graphene oxide functionalized LPG. Meanwhile, this sensing system provided a higher FOM compared to other FBG sensors due to a significantly smaller FWHM of the cladding mode (~0.2 nm). Compared to the SPR-TFBG technique, this proposed sensing system also offers a simpler solution for locating the wavelength based on intensity demodulation. In summary, the proposed BP-WTFBG sensor shows a significantly enhanced sensitivity at the low-RI region and provides a low-cost, harmless (to human body), and environmentally-friendly solution for biochemical sensing.

## Figures and Tables

**Figure 1 nanomaterials-10-01423-f001:**
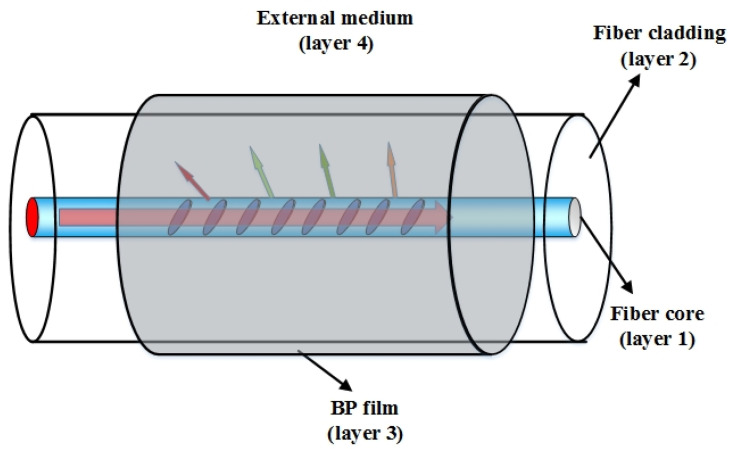
Schematic illustration black phosphorus (BP) coated weakly tilted fiber Bragg grating (WTFBG).

**Figure 2 nanomaterials-10-01423-f002:**
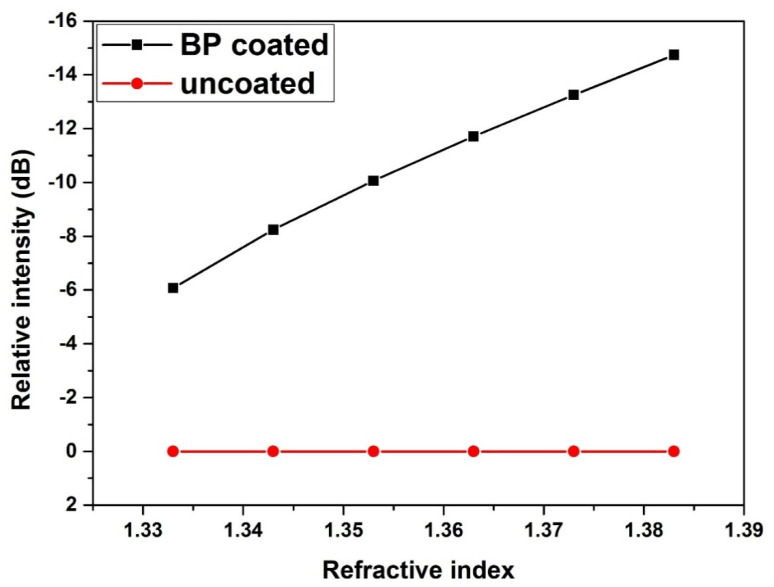
Simulation results of the relative transmission intensities of the uncoated and coated black phosphorus (BP) film structures for cladding modes of weakly tilted fiber Bragg grating (WTFBG) at 1552 nm with different refractive indices (RIs).

**Figure 3 nanomaterials-10-01423-f003:**
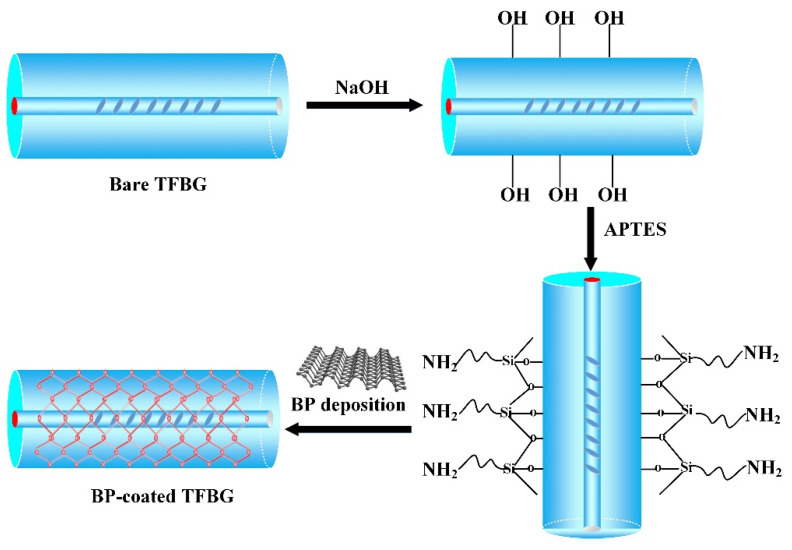
The flow diagram of the functionalized fiber surface and the deposition of black phosphorus (BP) onto weakly tilted fiber Bragg grating WTFBG.

**Figure 4 nanomaterials-10-01423-f004:**
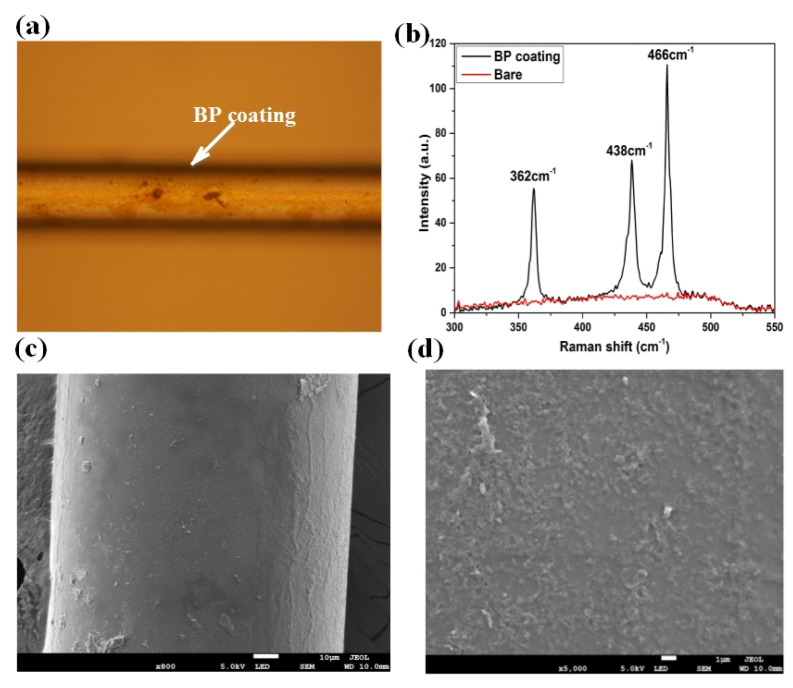
The (**a**) optical microscope image, (**b**) Raman spectrum, and scanning electron microscopy (SEM) pictures with (**c**) 800× and (**d**) 5000× magnification of the black phosphorus (BP) film coated weakly tilted fiber Bragg grating (WTFBG).

**Figure 5 nanomaterials-10-01423-f005:**
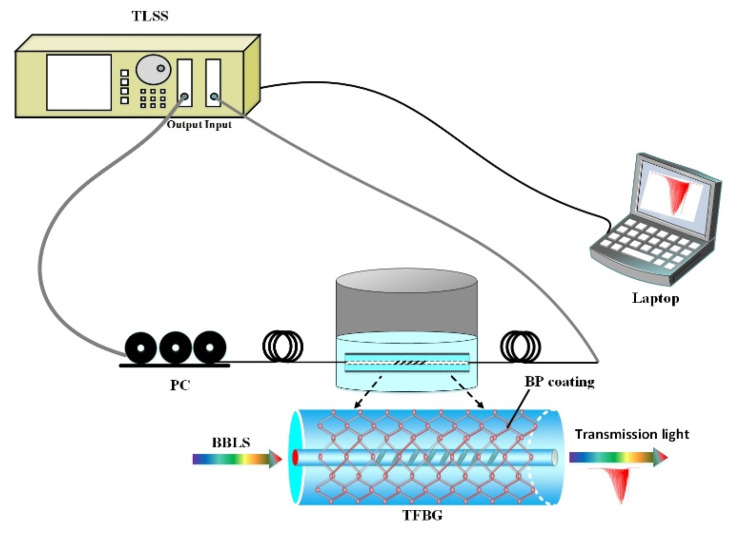
Experimental setup for the RI measurement; the insert shows the detailed structure of the black phosphorus integrated with weakly tilted fiber Bragg grating (BP-WTFBG).

**Figure 6 nanomaterials-10-01423-f006:**
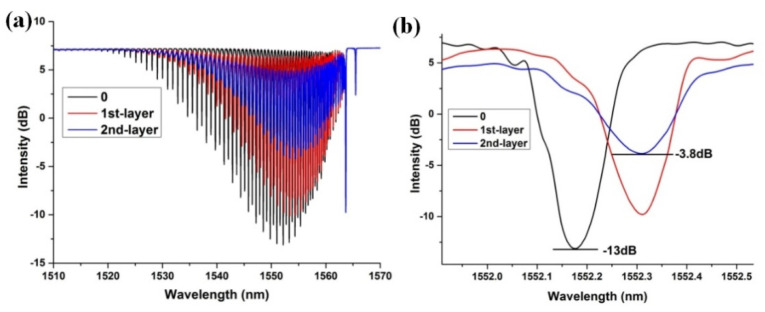
The transmission spectra of (**a**) whole cladding modes and (**b**) the cladding mode around 1552 nm of the black phosphorus integrated with weakly tilted fiber Bragg grating (BP-WTFBG) with the black phosphorus deposition process.

**Figure 7 nanomaterials-10-01423-f007:**
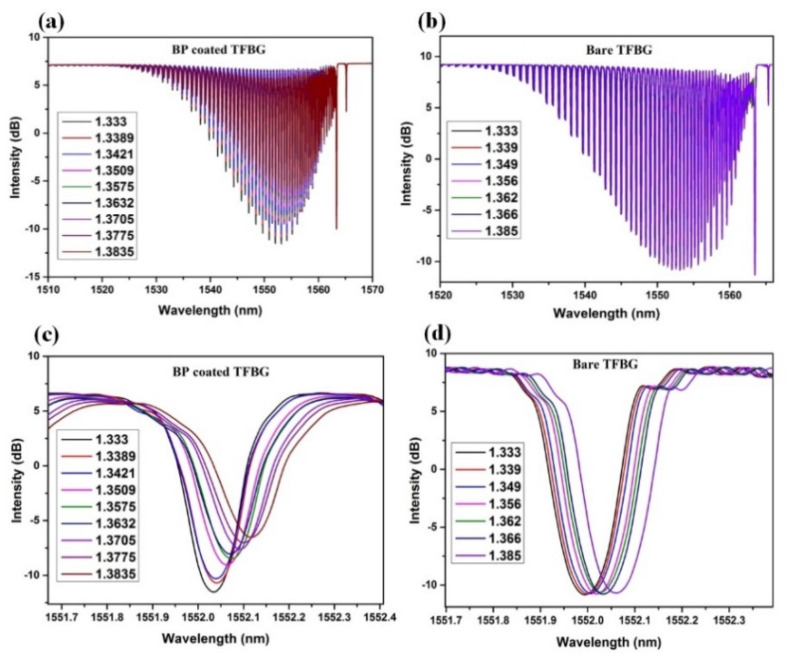
Transmission spectra of whole cladding modes and the cladding mode of around 1552nm for weakly tilted fiber Bragg grating (WTFBG) with (**a**,**c**), and without (**b**,**d**) black phosphorus (BP) coating, in the detection of solution with different refractive indexs (RIs).

**Figure 8 nanomaterials-10-01423-f008:**
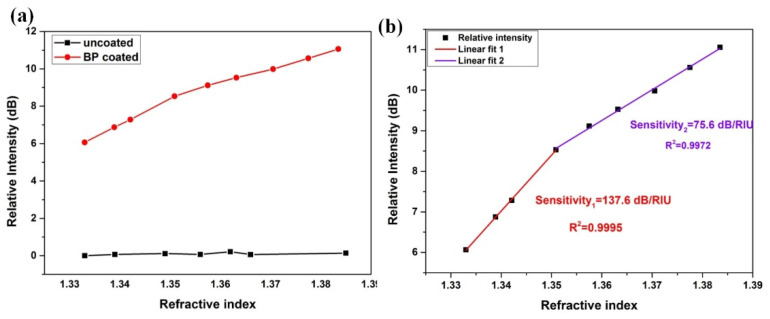
(**a**) The relative intensity for the minima of deepest cladding mode around 1552 nm of bare or black phosphorus integrated with weakly tilted fiber Bragg grating (BP-WTFBG) in the 1.33–1.38 RI region. (**b**) The linear fit for the relative intensity of BP-WTFBG against different RIs.

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
