# Peer review of "Refractometric Sensitivity Enhancement of Weakly Tilted Fiber Bragg Grating Integrated with Black Phosphorus"

_nanomaterials, 2020, doi:10.3390/nano10071423_

Round 1
Reviewer 1 Report
In this paper, the authors present the modification of an optical fiber with a layer of black phosphorous nanosheets with the aim to develop a refractive index sensor fiber. The study combines a theoretical part with other experimental. The methodology followed is convincing and the synthesis is well described. However, the characterization is poor, I miss information of each step, such as concentration of the silane groups or the properties of each BF layer. In the study should be included the effect in the response of each treatment and not only bare TFBG, for instance include if the monolayer of silane has influence in the optical properties. On the other hand, also appeared two different refractive index regions, this still require more discussion.
Minor changes:
- Improve the typing of equations 1, 2, 3 and 4.
- Improve Figure 1
Therefore, I think the paper still requires major changes to be publishable in this journal
Author Response
Thanks for your careful reading of our manuscript and your comments that are very helpful to improve the quality of this paper. I have alredy answered you problems or doubts. The detail is showed in the attachment below. Please to check it.

Reviewer 2 Report
I quite like the idea of this manuscript, which is to use black phosphorous integrated with weakly tilted fiber Bragg grating as a mechanism of novel sensor development. I recommend that the manuscript be accepted for publication after the following, relatively minor issues are addressed:
- This manuscript overall suffers from a variety of syntax-based errors that make it difficult to focus exclusively on the scientific content of the manuscript. A detailed proofreading is recommended.
- In the introduction, the authors write that refractive index sensing is booming because of advantages that include “compacted size.” The correct phrase is “compact size.” Moreover, additional information about how refractive index sensing (a technique) has a size advantage is requested in order to clarify the information provided.
- The authors refer to “black phosphorous” as “one of the famous two demission materials.” The phrase “two demission materials” should be clearly defined, and references to support the assertion that black phosphorous is “famous” should be provided.
- In the heading for section 2.1, the word “theory” is misspelled. This should be corrected.
- There is also a spelling error in the title for section 3.1. The correct spelling is “Experimental setup.”
- In general, this manuscript would benefit from a more detailed rationale of how and why certain features of the experiment were selected – i.e. why this kind/level of theory? How does it compare to experimental results? Why this kind of black phosphorous? Why this kind of layer by layer deposition technique? What are the advantages/challenges of each part of the experimental design selected? Including answers to these questions in this manuscript would dramatically enhance the scientific quality/depth of the results reported herein. It is strongly recommended that the authors make the requisite changes before the manuscript is accepted for publication.
Author Response

(The authors gave the same response as above.)

Reviewer 3 Report
The manuscript "Refractometric sensitivity enhancement of weakly tilted fiber Bragg grating integrated with black phosphorus", written by Zhang et al. describes a considerably interesting topic of BP coated Bragg grating. The theme is potentially sound. However, I believe the material is not very well described and on what materials were the results obtained. BP is an unstable system. Chemical structure of the bounded BP should be better described from the points of view of: a) Raman spectra should be used for a better description of the BP on the surface. Was the BP oxidized and how. How many layers of BP are present in average? Raman mapping should be performed to show a distribution of the material on the surface. b) XPS or IR spectroscopy should be added to better describe the chemical state of the nanomaterial. Minor comments: Equations 1 and 2 are not well displayed.
Author Response
Thanks for your careful reading of our manuscript and your comments that are very helpful to improve the quality of this paper. I have alredy answered you problems or doubts. The detail is showed in the attachment below. please to check it.

Round 2
Reviewer 1 Report
In the present revision, the authors have properly discussed my main concerns, therefore, I recommend the manuscript for publication in this journal.
Reviewer 3 Report
The text of the manuscript was considerably improved and many new results were added. I believe the manuscript can now be accepted for publication.